# Relationship of Glucose, C-peptide, Leptin, and BDNF in Maternal and Umbilical Vein Blood in Type-1 Diabetes

**DOI:** 10.3390/nu15030600

**Published:** 2023-01-24

**Authors:** Josip Delmis, Slavko Oreskovic, Vesna Elvedji Gasparovic, Mirta Starcevic, Mislav Herman, Nada Dessardo, Vito Starcevic, Marina Ivanisevic

**Affiliations:** Department of Obstetrics and Gynecology, University Hospital Medical Centre Zagreb, School of Medicine, University of Zagreb, 10000 Zagreb, Croatia

**Keywords:** BDNF, CRP, C-peptide, leptin, pregnancy, skinfold thickness, thyroid hormones, type-1 diabetes mellitus

## Abstract

The study aimed to determine the relationship between glucose, C-peptide, brain-derived neurotrophic factor (BDNF), and leptin between mother and fetus and neonatal weight. Methods: In the prospective observational cohort study, we included 66 women with type-1 diabetes mellitus (T1DM). According to the z-score for neonatal weight, patients were divided into healthy-weight neonates (*n* = 42) and overweight neonates (*n* = 24). The maternal blood samples were taken during pregnancy and cesarean section when the umbilical vein blood sample was also withdrawn. The maternal vein sera were analyzed for fasting glucose, C-reactive protein (CRP), leptin, BDNF, TSH, FT3, and FT4. The umbilical vein sera were analyzed for glucose, C-peptide, leptin, TSH, thyroid-stimulating protein (FT3), free thyroxine (FT4), and BDNF concentration. The neonatologist measured the skinfold thickness on the third day of neonatal life. Results: A strong correlation was confirmed between maternal and umbilical vein glucose concentration and maternal glucose and C-peptide in umbilical vein blood. A negative correlation was found between the concentration of BDNF in the umbilical vein and glucose in maternal blood. A strong correlation was seen between BMI and maternal blood leptin concentration, neonatal fat body mass, and umbilical vein blood leptin concentration. Higher BMI elevated BDNF, and TSH increase the odds for overweight neonates in the first trimester of pregnancy. Maternal higher leptin concentration in the first trimester decrease the odds of overweight neonates. Conclusions: Maternal glucose concentrations affect the fetus’s glucose, C-peptide, and BDNF concentrations. Leptin levels increase in maternal blood due to increased body mass index, and in the neonate, fat body mass is responsible for increased leptin concentrations.

## 1. Introduction

Type-1 diabetes (T1DM) is an autoimmune disease due to selective β-cell destruction and insulin secretion defects [1,2]. Diabetes in pregnancy creates numerous problems for both mother and child [3,4]. Poor glycemic control in pregnant women with T1DM increases the risk of miscarriage, preterm birth, hypertension/preeclampsia, maternal and perinatal morbidity, and mortality [5,6]. An intensive approach to reaching normoglycemia in the preconception period and during pregnancy is mandatory to achieve a successful pregnancy outcome and a healthy newborn. Adequate nutrition and intensified insulin therapy lead to the desired optimal glucose control.

Glucose, amino acids, and fatty acids are essential for fetal growth. Glucose crosses the placenta by facilitated diffusion. Fetal glucose supply depends on several factors, such as uteroplacental flow, maternal blood glucose concentration, placental metabolism, and glucose transporter activity [7]. Glucose transporters (GLUTs), located in the syncytiotrophoblast and basement membrane of the chorionic villi, mediate glucose transport across the placenta [8].

Type-1 diabetes is associated with changes in the content and composition of maternal lipids [9]. Maternal sources of fetal lipids include lipoprotein-associated triacylglycerol, phospholipids, cholesterol esters, and free fatty acids [10]. The current understanding of maternal lipid supply for the fetus involves the transfer of free fatty acids and free cholesterol [10]. The fatty acids participate in membrane phospholipid synthesis and play a substantial role in growth and development [11]. Macrosomic neonates delivered by women with T1DM had a higher ponderal index and higher concentrations of insulin, leptin, and FAs in the umbilical vein and artery compared to control group newborns [12]. An imbalance in fatty acid intake during pregnancy may result in permanent changes in the fetal hypothalamus development and appetite control later in life, neuroendocrine function, and energy metabolism, leading to altered metabolic programming [11].

The fetus requires amino acids for protein synthesis, carbon accretion, oxidative metabolism, and biosynthesis, which ultimately determine the growth rate in utero [13]. The fetal supply of amino acids depends on the placenta’s transport capacity [13]. The concentration of most amino acids is higher in fetal plasma than maternal plasma, indicating that they are actively accumulated across the syncytiotrophoblast, the transporting and hormone-producing epithelium of the placenta [13].

According to Pedersen’s hypothesis that fetuses of mothers with T1DM exhibit accelerated growth, increased maternal glucose transmission from the placenta to the fetus results in fetal hyperinsulinemia [14]. Hyperglycemia and hyperlipidemia in pregnant women with T1DM boost the fetal supply of glucose and fatty acids, causing accelerated fetal growth [9].

The fetus can synthesize fatty acids de novo using glucose as the precursor to form triacylglycerols and to store them in fat depots [10]. Newborns born of a mother with T1DM have increased fat depot. Classic diabetes fetopathy is characterized by macrosomia, increased adipose tissue and glycogen, and a plethoric cushingoid appearance.

Thyroid disorders and diabetes mellitus are common chronic endocrine disorders that often coexist [15]. Common autoimmune and non-autoimmune causes can lead to hypofunction or hyperfunction of more endocrine glands [15]. Pregnant women with subclinical hypothyroidism during pregnancy are more prone to glucose and lipid metabolism disorders [15].

Brain-derived neurotrophic factor (BDNF) is a member of the nerve growth factor (NGF) family [16,17]. It exerts its biological function through the tropomyosin-related kinase B (TrkB) receptor [16]. BDNF plays a role in neurogenesis, regulates synaptic transmission, maintains adult synapses in the CNS, and improves cognitive function [16]. It is synthesized in the central nervous system during fetal development [17]. BDNF plays a significant role in brain development and function and glucose and lipid metabolism [18]. BDNF stimulates angiogenesis, placental development, and fetal growth [19]. BDNF is found in the brain and the periphery [20]. It regulates glucose and energy metabolism and prevents β-cell depletion.

Neonates of mothers with T1DM are often macrosomic and obese. Leptin levels were associated with higher body weight and lower BDNF levels [21]. There is evidence that the placenta also produces leptin, not just maternal adipose tissue, a significant contributor to the increase in maternal leptin concentration [22]. Leptin participates in the regulation of body weight through the suppression of appetite and stimulation of energy consumption [21].

C-reactive protein (CRP) is a clinically useful marker of systematic inflammation, and its synthesis occurs in the liver and is induced by infection and tissue injury. Higher levels of CRP in T1DM pregnancy were positively associated with preeclampsia [23]. CRP is a crucial plasma protein that can bind leptin [24].

The study aimed to determine the relationship between glucose, C-peptide, BDNF, and leptin between mother and fetus. The secondary endpoint of the research was to analyze the impact of maternal age, duration of type-1 diabetes mellitus, BMI, HbA1c, TSH, leptin, and BDNF on neonatal overweight.

## 2. Materials and Methods

### 2.1. Ethical Statements

The Ethics Committee School of Medicine at the University of Zagreb approved the study (No. 380-59-10106-19-111/26) within the scientific project PRE-HYPO No. IP-2018-01-1284. All women in the study provided informed consent for themselves and their newborns.

### 2.2. Study Participants

In the prospective study, we consecutively included 66 women with type-1 diabetes mellitus before completing 10 gestational weeks with a single living fetus during the study period from 1 February 2019 to 31 January 2021. All participants were admitted to the Department of Obstetrics and Gynecology at least once or repeatedly in each trimester. The daily glucose profiles of patients with T1DM were determined, and plasma glucose (9/day) was monitored for 2–3 days. None of the pregnant participants in this study experienced a hypoglycemic coma or needed third-party assistance during hypoglycemia or glucagon/intravenous glucose during the hypoglycemic event. No episodes of severe hypoglycemia were reported for the whole pregnancy.

#### 2.2.1. Inclusion Criteria

We included 66 pregnant women with type-1 diabetes and singleton pregnancies who received insulin therapy for at least two years. At pregnancy confirmation, the HbA1c was ≤8% (64 mmol/mol). All pregnant women received intensified insulin therapy with fast-acting insulin and long-acting insulin.

According to the z-score for neonatal weight, the participants were divided into healthy-weight neonates (*n* = 44) and overweight neonates (*n* = 24).

#### 2.2.2. Exclusion Criteria

Pregnant women with T1DM who had proliferative retinopathy, nephropathy, and chronic hypertension were excluded from the study. Exclusion criteria were a maternal age of fewer than 18 years and a major fetal defect observed on the scan performed at 11 to 13 weeks of gestation.

### 2.3. Data Collection

Pregnancy information, gestational weight gain expressed as the difference in weight before pregnancy (self-reported) and at the time of delivery, and pre-pregnancy body mass index (kg/m^2^) were calculated from the pre-pregnancy values and collected for each participant. Neonatal macrosomia was defined as a birth weight of ≥4000 g. Large for gestational age (LGA) was defined if birth weight was >90th percentile for gestation weight and sex [25]. Birth weight z-score specific for sex and gestational age were calculated according to the growth curves published by WHO in 2006 [26]. The z-score for neonatal weight provides information on underweight (percentile < 2), healthy-weight (percentile ≥ 2 < 98), and overweight (≥98) neonates.

The prospective study included newborns of T1DM mothers. The newborns had been evaluated for their clinical condition immediately after birth and monitored for somatic and neurological outcomes. Anthropometric newborn measurements (body mass and length, abdominal circumference, head circumference) were taken upon birth. Standardized measuring instruments were used for anthropometric measurements (scale for weight, length meter, caliper). On the third day of neonatal life, skinfold thickness was measured for subscapular, abdominal, triceps, biceps, and upper leg performed using a Harpenden skinfold caliper with a calibration dowel [27]. To calculate fatty mass percentage, we used the equations of Slaughter et al. [28].

Maternal blood samples were taken during pregnancy and cesarean section. In each trimester, the maternal vein sera/whole blood was analyzed for glucose, BDNF, leptin, C-peptide, CRP, thyroid hormones (TSH, FT3, and FT4), and HbA1c percentage.

Umbilical vein blood samples were obtained immediately after birth but before removing the placenta through puncture of the umbilical vein for glucose, C-peptide BDNF, TSH, FT3, FT4, and leptin. Maternal venous blood samples were collected during delivery as well. Neonatal birth weight (g), length (cm), and the 1 min and 5-min Apgar scores were measured postnatally.

### 2.4. Blood Sample Analyses

The hexokinase method quantified glucose levels on a Cobas C301 analyzer with reagents from the same manufacturer (Roche, Basel, Switzerland). The HbA1c levels in whole blood were measured by turbimetric inhibition immunoassays on a Cobas C501 instrument (Roche, Basel, Switzerland). The serum C-peptide concentration was evaluated by electrochemiluminescence immunoassays (ECLIAs) with Elecsys immunoassays/Roche Diagnostics, Switzerland) with the lower detection limit of 0.003 nmol/L.

According to a homeostasis model assessment, neonatal insulin resistance was calculated using online software (https://homa-calculator.informer.com/2.2/, accessed on 23 January 2023).

The ARCHITECT TSH, FT3, and FT4, a chemiluminescent microparticle immunoassay (CMIA), were used for the quantitative determination of human thyroid stimulating hormone (TSH), free triiodothyronine (FT3), and free thyroxine (FT4) in human serum and plasma (Abbott Ireland Diagnostics, Longford Co. Longford Ireland). Subclinical hypothyroidism is defined based on the TSH value. If the TSH value was less than 2.5 mIU/L, we considered euthyroidism for the first trimesters. Subclinical hypothyroidism is diagnosed based on a TSH value between 2.5–5.0 mIU/L).

Sandwich ELISA Kit was used to determine BDNF serum concentration, ChemiKine, Merck KGaA, Darmstadt, Germany (No. Cyt306). Test sensitivity is 15 pg/mL.

Leptin serum concentration was determined by sandwich Kit, Tecan, IBL International, Hamburg, Germany (Cat. No. MD53001).

Automated CRP measurements were performed with an Abbot Alinity C analyzer (Abbot Laboratories Wiesbaden, Germany). The declared detection limit was 0.03 mg/L (Immunoturbimetry method).

### 2.5. Sample Size

We tested the correlation coefficient, which was predicted at 0.60 for power calculations between maternal and umbilical vein serum glucose concentration. For a sample size of a total of 34 patients, the α-level is 0.05 and the power is 90%.

### 2.6. Statistical Analyses

Statistical analyses were performed using the statistical package SPSS version 26 (IBM, Armonk, NY, USA). Absolute and relative frequencies represent categorical data. A Pearson Chi-square test was used to test group differences between categorical variables. Numerical data are described by the mean (arithmetic mean) and standard deviation in the distribution following normal, and in other cases, by median and interquartile range limits. We used one-way ANOVA and Tukey’s test for multiple analysis tests. The significance level was set at α < 0.05. Student’s *t*-test tested group differences between normally distributed continuous variables, and differences between nonnormally distributed continuous variables were tested by the Mann–Whitney U test. Pearson’s correlation coefficient was used to evaluate the correlation between normally distributed numerical variables. The not normally distributed data were log-transformed (natural logarithm) before analyses. Regression with the Spearman correlation coefficient (r_rho_) was performed for non-normal data. For repeated measurement of continuous data, the Wilcoxon signed-rank test was used. All *p* values are two-sided. The significance level was set at *p* < 0.01.

## 3. Results

### 3.1. Demographic, Obstetric, and Neonatal Data

Maternal characteristics taken into account were age, duration and years of onset of diabetes, height, weight, BMI, parity, duration of pregnancy, HbA1c, and mean glucose by trimester of pregnancy and during delivery. The number of pregnant women in the first trimester with HbA1c ≥ 6.6% is 34, and the number with ≤6.5% is 32. In the second trimester of pregnancy number of HbA1c ≥ 6.6% significantly decreased (*p* < 0.001). We measured glucose in capillary plasma at three-hour intervals, starting at 7 am and continuing at 10 am, 1 pm, 4 pm, 7 pm, 10 pm, 1 am, 4 am, and 7 am on the second day. Mean glucose values were calculated based on the daily glucose profile in capillary plasma. In the first trimester, the mean value with SD of glucose was 5.8 ± 1.4, in the second trimester it was 5.3 ± 1.2, and in the third trimester it was 5.5 ± 1.0. Comparing mean glucose values in the first and third trimesters, no significant decrease in mean glucose values was found.

The number of overweight neonates was 22, and the number of healthy-weight neonates was 44 (Table 1).

Maternal demographic, obstetrics, and laboratory data are shown in Table 1. The number, lowest, highest, and mean values with standard deviation or median and interquartile range (IQR) are shown.

Patient characteristics were age 30.5 ± 5.3 years, duration of T1DM 14.3 ± 9.4 years, years of onset diabetes 16.2 ± 9.2 years, and BMI before pregnancy 23.9 ± 4.8 kg/m^2^.

Obstetrical characteristics were nulliparous 57.6% (38/66), gestational age at delivery 38.1 ± 0.9 weeks, gestational weight gain 13.0 ± 4.6 kg, and BMI at delivery 28.4 ± 4.7 kg/m^2^.

Table 1 shows the decrease in HbA1c and BDNF concentration from the first trimester to delivery (*p* < 0.001) and the increase in leptin and CRP concentration from the first trimester to delivery (*p* < 0.001) in mothers. Significantly higher concentrations of BDNF in maternal than in umbilical vein serum (*p* < 0.001) are presented in the table.

#### 3.1.1. Neonatal Clinical and Laboratory Data

Neonatal characteristics analyzed were the number of male and female newborns, birth weight and length, z-score of neonatal weight and percentile of neonatal z-score, head and abdominal circumference, skinfold thickness, Apgar index in the 1st to 5th minute, pH, glucose and C-peptide, BDNF, leptin concentration, and IR (HOMA 2), TSH, FT3, and FT4 in umbilical vein blood. Neonatal clinical and laboratory data are shown in Table 2. The number, lowest, highest, and mean values with standard deviation or median and interquartile range (IQR) are shown.

The number of males was 39 (43.9%). Prevalence of neonatal overweight was 36.4% (24/66), preterm delivery occurred in 10.6% (7/66) of participants, mean birth weight was 3614 ± 584.0 g, mean birth length was 49.6 ± 2.3 cm, mean head circumference was 34.8 ± 1.5 cm, mean abdominal circumference was 33.3 ± 2.7 cm, the z-score of neonatal weight was 0.9 (0.4–1.6), the sum of skinfold thickness was 25.6 ± 3.8 cm, fat body mass was 459.3 ± 119.9 g, lean body mass was 3155.4 ± 484.2 g, and mean pH value in umbilical blood at birth was 7.21 ± 0.08. The following mean values at birth were measured in the umbilical vein: TSH 4.9 (3.7–6.6) mIU/L, FT3 2.6 ± 0.6 pmol/L, FT4 13.0 ± 1.3 pmol/L, C-peptide 820 (570–1380), BDNF 304.7 (231.6–458.3), and leptin 13.5 (8.9–27.1) ng/L. More detailed information on the skin fold thickness measured in neonates three days after delivery and calculated lean and fat body mass are presented in Table 2.

The mothers of the overweight group neonates were younger than mothers who gave birth to the healthy-weight neonates. A higher number of pregnant women with HbA1c ≥ 6.6 in the group with overweight than healthy-weight neonates (χ^2^ test, *p* = 0.023) was found. These data are presented in more detail in Table 3.

The overweight neonates, in addition to being heavier, had a larger abdominal circumference, thicker skinfolds, higher T4 concentration, and a higher maternal–umbilical vein glucose ratio (Table 4).

#### 3.1.2. Linear Regression of Glucose Concentration between Maternal and Umbilical Vein Serum

Immediately before the child’s extraction at cesarean section, we withdraw the maternal venous blood. Instantly after the child’s birth, we took the umbilical vein blood sample while the placenta was still in situ.

Figure 1 shows the linear regression between maternal and fetal glucose concentration.

A significantly higher maternal and umbilical vein glucose concentration ratio was found in the overweight newborn group (Figure 2).

### 3.2. Nonparametric Correlation between Glucose Concentration in Maternal Vein and C-peptide Concentration in Umbilical Vein

Significant nonparametric correlations were found between maternal venous serum and umbilical vein C-peptide concentrations (r_rho_ = 0.566, *p* < 0.001) (Figure 3). The difference between maternal and umbilical glucose concentration was significantly higher in overweight than in healthy-weight newborns (1.34 (1.14–1.54): 1.04 (0.93–1.38), *p* = 0.012).

Significant nonparametric correlations were found between umbilical vein serum glucose and C-peptide concentration (Figure 4). A higher ratio of umbilical vein glucose and C-peptide was found in healthy-weight 5.3 (4.2–7.2) than in overweight neonates 3.9 (2.9–5.9).

### 3.3. Nonparametric Correlation between BMI, Neonatal Weight, and Leptin Concentration

Significant nonparametric correlations were demonstrated between maternal BMI and leptin concentration at the time of delivery (r_rho_ = 0.501, *p* < 0.001) (Figure 5).

Significant nonparametric correlations were found between the z-score of neonatal weight and umbilical venous serum leptin concentration (r_rho_ = 0.316, *p* = 0.014) (Figure 6).

Significant nonparametric correlations were found between maternal and umbilical vein venous serum BDNF concentration (r_rho_ = 0.494, *p* < 0.001) at the time of delivery (Figure 7).

Figure 8 shows lean body mass results in male and female newborns. Female newborns did not have a significantly higher percentage of fat mass, nor the amount of fat mass, but females had lower lean body mass than male neonates (3046.2 ± 465.2: 3294.8 ± 479.6, *p* = 0.037).

Significant correlations were found between birthweight and umbilical vein venous serum FT4 concentration (r = 0.320, *p* = 0.024) (Figure 9).

A significantly higher umbilical vein FT4 concentration ratio was found in the overweight newborn group (Figure 10).

BMI, BDNF, and TSH increased odds for overweight neonates in the first trimester. Leptin concentration in the first trimesters decreased the odds for overweight neonates (Table 5).

## 4. Discussion

### 4.1. Excessive Fetal Growth despite Optimal Glycemic Control

In this prospective study, the impact of optimal glycemic control on neonatal weight in pregnant women with type-1 diabetes mellitus is shown. Mean HbA1c values decreased from 6.8 to 6.0%, and glucose ranged from 5.8 to 5.5 mmol/L from the first to the third trimester. Despite optimal glycemic control, 25.8% of neonatal macrosomia and 36.4% of overweight neonates were born to pregnant women with T1DM. Even the recurrent episodes of hypoglycemia in pregnant diabetic women may result in excessive fetal growth caused by fetal hyperinsulinism [29]. An explanation for this can be found in the article by Desoye, G. and Nolan, C.J., “The fetal glucose steal: An underappreciated phenomenon in diabetic pregnancy” [29]. The authors explain that accelerated fetal growth could result from the early establishment of fetal hyperinsulinemia. Fetal pancreatic beta cells release insulin in the first trimester of pregnancy [30]. Maternal diabetes is associated with fetal hyperinsulinemia; however, diabetes is also a disorder of lipid metabolism, which may be important for the development of fetal hyperinsulinemia. Early-onset neonatal hyperinsulinemia accelerates glucose clearance into fetal tissues [29]. Even during normal maternal glucose levels, fetal hyperinsulinemia will still lower fetal glucose concentration, resulting in accelerated fetal growth. It is highly conceivable that fetal hyperinsulinemia could also drive a fetal fatty acid and/or an amino acid steal [29].

Glucose is a significant source of energy for the placenta and fetus. The fetus is dependent on maternal glucose because it synthesizes it in small amounts [31]. Simultaneous maternal and umbilical vein blood sampling during cesarean section found a strong correlation between maternal and fetal glucose concentrations. Maternal hyperglycemia is the cause of increased fetal glucose, which then stimulates the beta cells of the fetal pancreas to produce insulin. In this study, we found a higher maternal and umbilical vein glucose concentration ratio and a lower ratio of umbilical glucose and C-peptide in the overweight neonate group. Desoye, G. and Nolan, C.J. [29] consider that fetal hyperinsulinemia, through its effect on lowering fetal glycemia, increases the glucose concentration gradient across the placenta and, consequently, the glucose flux to the fetus, which was confirmed in this study. According to Pedersen’s hypothesis that fetuses of mothers with T1DM exhibit accelerated growth, increased maternal glucose transmission from the placenta to the fetus results in fetal hyperinsulinemia [14].

We have also shown that the cord blood levels of C-peptide, leptin, and insulin resistance were higher in the T1DM group compared with the healthy controls. T1DM is a known factor that predisposes women to have hypertrophic newborns through an enhanced fetal glucose–insulin axis. The fetus can synthesize fatty acids de novo using glucose as the precursor to form triacylglycerols and to store them in fat depots [10]. Our previous studies showed higher glucose and fatty acid concentrations in the blood of diabetic mothers and umbilical veins compared to non-diabetic mothers and their fetuses [9]. Our findings eventually lead to more fetal adipose tissue, as reflected by higher leptin levels.

The skinfold thickness was significantly higher in overweight than in healthy-weight neonates. Overweight neonates had higher fat body mass than healthy-weight neonates, which resulted from an increased transfer of glucose and free fatty acids across the placenta and fetal hyperinsulinemia. Male newborns are heavier than females, but no difference was found in the fat mass percentage. Female neonates had significantly lower lean body mass than male neonates, which is in line with the research of other authors [32]. Birthweight and length are associated with umbilical vein C-peptide (r_rho_ = 0.418 *p* = 0.002; r_rho_ = 0.392, *p* = 0.004). Neonatal fat body mass was associated with leptin (r_rho_ = 0.343, *p* = 0.007).

### 4.2. Leptin in Maternal and Fetal Blood

Leptin levels increase during pregnancy and strongly correlate with maternal weight and BMI. They rise in pregnancy due to increased adipose tissue and leptin production in the placenta. Leptin in the umbilical vein correlates with neonatal fat body mass. Maternal leptin levels were significantly higher than fetal circulation. No correlation was found between maternal and fetal leptin, which is evidence that maternal leptin is not transmitted from the mother across the placenta to the fetus. Fetal hyperinsulinemia stimulates fat deposition in adipocytes, increases the proportion of adipose tissue, and accelerates fetal growth. Thus, leptin is a biomarker of fetal obesity.

Higgins, M.F. et al. [33] found significantly higher values of fetal leptin in pregnant women with T1DM and T2DM compared to non-diabetic mothers. The authors speculate that newborns of diabetic mothers have leptin resistance, which leads to the inevitable metabolic repercussions later in life. This is why strict glycemic control during pregnancy is essential. C-peptide and leptin levels were higher in overweight than in eutrophic newborns.

### 4.3. CRP in T1DM Pregnant Women

This prospective study showed a significant increase in CRP from the first to the third trimester. We showed significant correlations between pre-pregnancy BMI and CRP in the first trimester of pregnancy, which agrees with other authors [34]. We also established a strong correlation between leptin concentration and CRP in the first trimester of pregnancy; other authors showed similar results [34]. Although the increase in CRP and plasma leptin concentration has been observed in obesity, it is unclear whether elevation in CRP is due to acute inflammation, adipose tissue expansion, or both.

### 4.4. A BDNF during Pregnancy

This study showed a significant decrease in BDNF levels during pregnancy in mothers with T1DM. We found an inverse correlation (r_rho_ = −0.393, *p* = 0.001) between the age of pregnant women and BDNF. Younger pregnant women had higher BDNF concentration than older. Other studies have also shown a decrease in BDNF during pregnancy in healthy and pathological conditions, in maternal depression in pregnancy, in pregnant women with T1DM, and preeclampsia [18,35]. Christian, I.M. et al. [36] found significantly lower serum BDNF in pregnant versus non-pregnant women. D’Souza, V. et al. [18] determined BDNF levels during pregnancy in pregnant women with preeclampsia and healthy pregnant women. They found a decline in the concentration of BDNF with the duration of pregnancy in both groups, and a significantly lower concentration of BDNF was found in pregnant women with preeclampsia compared to healthy controls. The authors believe that BDNF plays an almost pivotal role in developing the maternal–fetal–placental unit during pregnancy. Decreased BDNF levels during pregnancy are due to abnormal placental development in preeclampsia; the authors conclude that it might involve placental development [18]. The concentration of BDNF in the maternal venous serum during childbirth is higher than the value of BDNF in the umbilical vein. Lower levels of BDNF in umbilical vein serum were found in preterm compared to term births (193.5 (128.6–290.3): 310.5 (251.2–466.2), *p* = 0.016). With the duration of pregnancy, the level of BDNF in fetal circulation increases. There is evidence that BDNF plays an important role in the proper growth, development, and plasticity of glutamatergic and GABAergic synapses, and through modulation of neuronal differentiation, it influences serotonergic and dopaminergic neurotransmission [35,37]. Due to the high correlation coefficient between the maternal and umbilical veins, the transition of BDNF from mother to fetus cannot be ruled out. The origin of circulating BDNF in the blood of the mother and fetus is also unclear. Kodomari I. et al. [38] demonstrated in mice that maternal BDNF reaches the fetal brain through the uteroplacental barrier. The authors injected recombinant BDNF into a pregnant mouse, and dose levels of BDNF increased in the fetal brain. These results suggest the transport of BDNF across the placenta [39].

Negative correlations were found between BDNF and glucose in maternal venous serum and between BDNF and glucose in umbilical vein serum. Our results showed that lower glucose concentration in diabetic pregnant women increases BDNF concentration. A recent finding suggests that BDNF regulates energy homeostasis [40]. BDNF treatment of diabetic animals resulted in decreased plasma glucose, non-esterified fat, and phospholipids [41]. It plays a substantial role in regulating food and energy balance in adults.

### 4.5. Subclinical Hypothyroidism in Type-1 Diabetic Pregnant Women

This study showed a high prevalence of subclinical hypothyroidism in pregnant women with T1DM. The diagnosis of subclinical hypothyroidism was established in the first trimester of pregnancy based on TSH values (TSH < 2.5–5.0 mIU/L). The occurrence of subclinical hypothyroidism was found in 50.8%. Hashimoto’s thyroiditis was diagnosed in 12 (18.2%) pregnant women before pregnancy. Thyroid hormones regulate the metabolism of glucose, lipids, and calcium in pregnant women and fetuses, and they are important for the normal growth and development of the fetus. Overweight fetuses had a higher concentration of FT4 in the umbilical vein. Associations between umbilical vein FT4 and z-score of neonatal weight suggest that fetal thyroid function may be important in regulating fetal growth, which is in line with Shields, B.M. et al. [42].

Pregnant women with subclinical hypothyroidism during pregnancy are more prone to glucose and lipid metabolism disorders [15]. It is important to determine thyroid hormones and thyroid peroxidase antibodies (TPO) in women with T1DM. All pregnant women with subclinical hypothyroidism in this study received levothyroxine.

The levels of TSH, BMI, and BDNF increased the odds of giving birth to overweight newborns. Surprisingly, maternal leptin concentration in the first trimesters decreased the odds of overweight neonates. Some studies have shown that mothers with a higher BMI give birth to heavier children, and mothers with higher leptin concentrations give birth to heavier children [43,44]. Catalano et al. [44] found an association between maternal leptin in late pregnancy and higher neonatal obesity. In the overweight group were 14 (58.3%) pregnant women with normal BMI and with lower leptin concentration of 9.8 (6.7–12.7), while in the healthy-weight group were 31 (72%) pregnant women with normal BMI and with higher leptin concentration of 14.3 (9.6–19.7), *p* = 0.049. Two obese pregnant women (8.3%) were in the overweight group, and 6 (14.3%) were in the healthy-weight group. When comparing the concentrations of leptin in the first trimester and at the time of cesarean section, a significant increase was found in pregnant women with normal weight (*p* = 0.004) and obesity (*p* = 0.043) but not in others i.e. mothers with overweight body mass index. Better glycemic control and less weight gain in obese pregnant women resulted in the higher prevalence of birth of healthy-weight newborns. These findings illustrate that multiple factors influenced fetal growth in pregnancy with type 1 diabetes, but the most important is early onset fetal hyperinsulinemia.

### 4.6. Strengths of the Study

This study investigated relationships between maternal serum glucose and glucose, C-peptide, BDNF, and leptin in umbilical vein serum. Maternal blood was obtained immediately before childbirth, and umbilical vein blood was obtained before the placenta was separated, reducing the influence of the postpartum period and allowing the results to represent the in vivo situation as closely as possible. As all women delivered by elective cesarean section were denied access to food, the levels of glucose, lipids, and other parameters measured in umbilical vein serum were not influenced by the mode of delivery or the nutritional status of the women. The combination of this measurement between the mother and the umbilical vein allows a conclusion on how maternal hyperglycemia and hypoglycemia affect the fetus.

The potential limitation of this study is the lack of a healthy control group.

## 5. Conclusions

Maternal glucose concentration affects the fetus’s glucose, C-peptide, and BDNF concentrations. Lower glucose levels in maternal and fetal blood increase the concentration of BDNF. Due to the high correlation between the mother and the umbilical vein, the transition of BDNF from mother to fetus cannot be ruled out. BMI and fat body mass affect maternal and neonatal leptin levels. BMI, BDNF, and TSH increased odds for overweight neonates in the first trimester of pregnancy, and maternal leptin concentration decreased the odds of overweight neonates.

Based on our results, the continuation of the study will assess whether BDNF, leptin, FT4, and C-peptide from the umbilical vein can predict children’s physical growth and cognitive development after one year of life.

## Figures and Tables

**Figure 1 nutrients-15-00600-f001:**
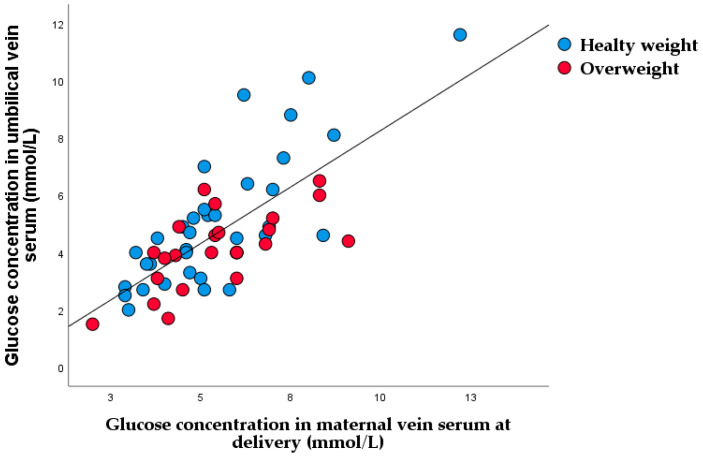
Linear regression of glucose concentration between maternal and umbilical vein serum (r = 0.752, *p* < 0.001).

**Figure 2 nutrients-15-00600-f002:**
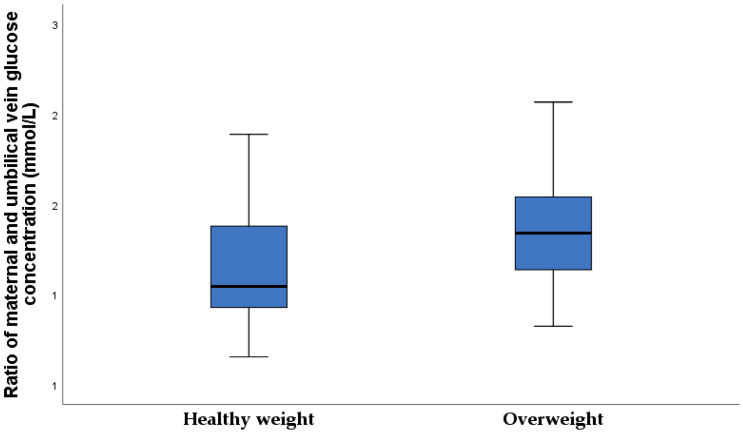
The maternal and umbilical vein glucose concentration ratio in two study groups (healthy-weight neonates 1.2 ± 0.3, overweight neonates 1.4 ± 0.7; *p* = 0.021).

**Figure 3 nutrients-15-00600-f003:**
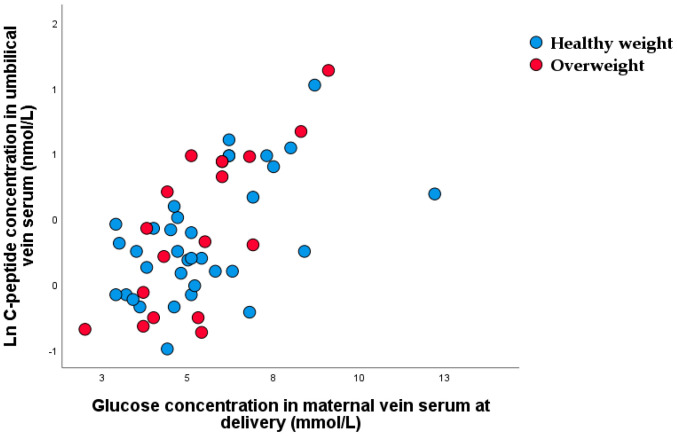
Scatter plot of nonparametric correlations between maternal venous glucose and umbilical vein C-peptide concentrations (r_rho_ = 0.566, *p* < 0.001).

**Figure 4 nutrients-15-00600-f004:**
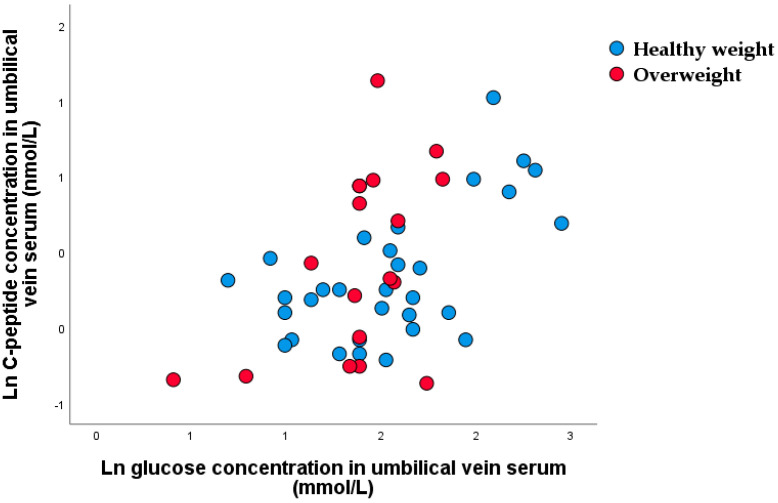
Scatter plot of nonparametric correlations between glucose and C-peptide concentrations in umbilical vein serum (r_rho_ = 0.449, *p* < 0.001).

**Figure 5 nutrients-15-00600-f005:**
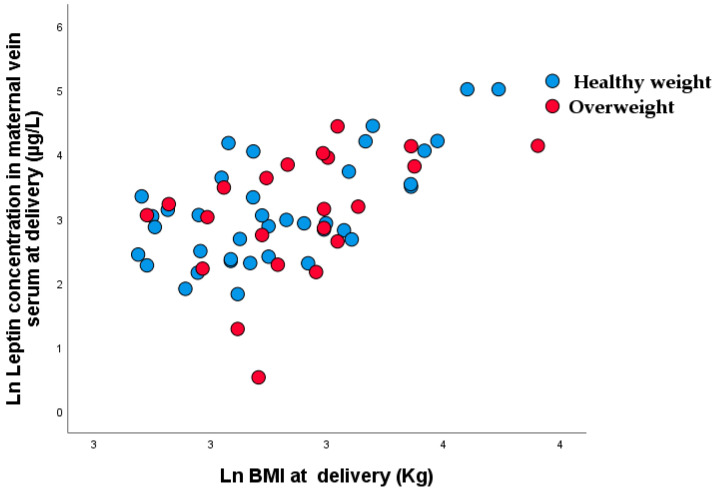
Scatter plot of nonparametric correlations between maternal body mass at the time of delivery and leptin concentrations at the time of delivery (r_rho_ = 0.501, *p* < 0.001).

**Figure 6 nutrients-15-00600-f006:**
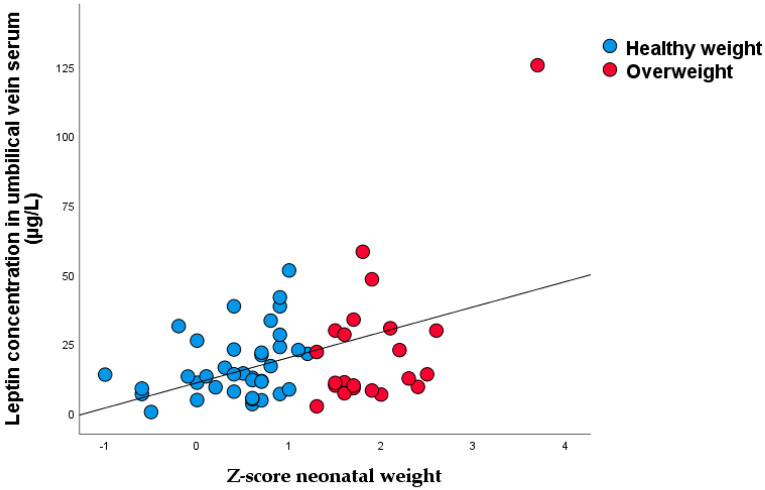
Linear regression of between z-score of neonatal weight and leptin in umbilical vein serum (r = 0.431, *p* = 0.001).

**Figure 7 nutrients-15-00600-f007:**
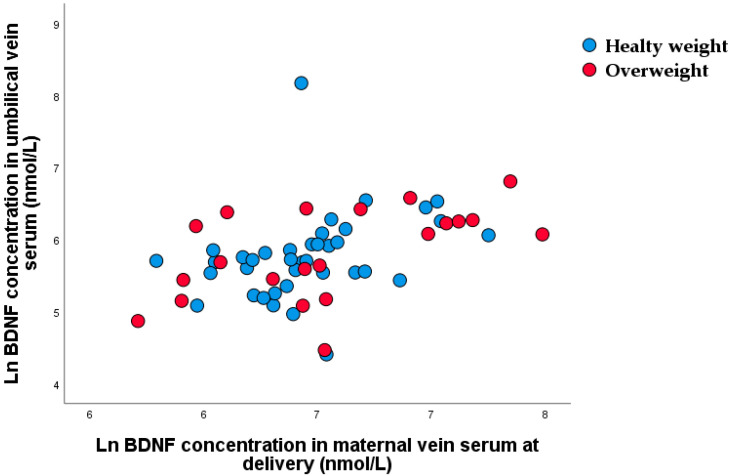
Scatter plot of nonparametric correlations between maternal and umbilical vein serum BDNF at the time of delivery (r_rho_ = 0.494, *p* < 0.001).

**Figure 8 nutrients-15-00600-f008:**
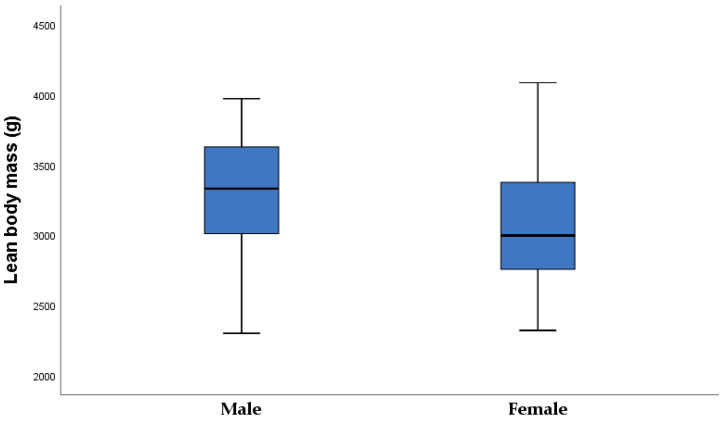
Lean body mass in male and female neonates (male 3294.8 ± 479.6: female 3046.2 ± 465.2, *p* = 0.037).

**Figure 9 nutrients-15-00600-f009:**
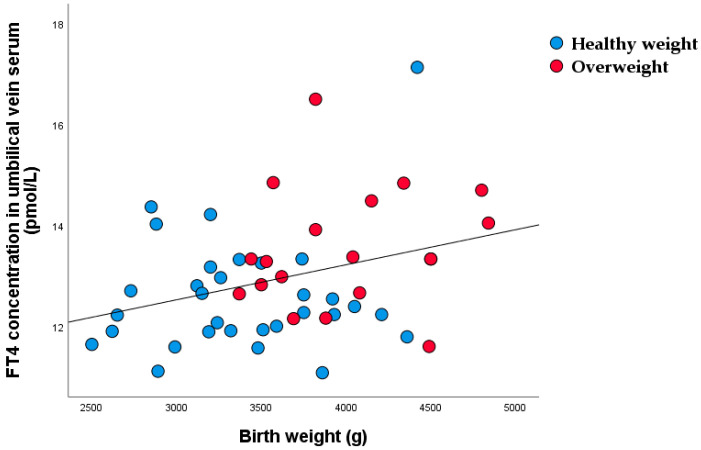
Linear regression between birth weight and FT4 in umbilical vein serum (r = 0.320, *p* = 0.024).

**Figure 10 nutrients-15-00600-f010:**
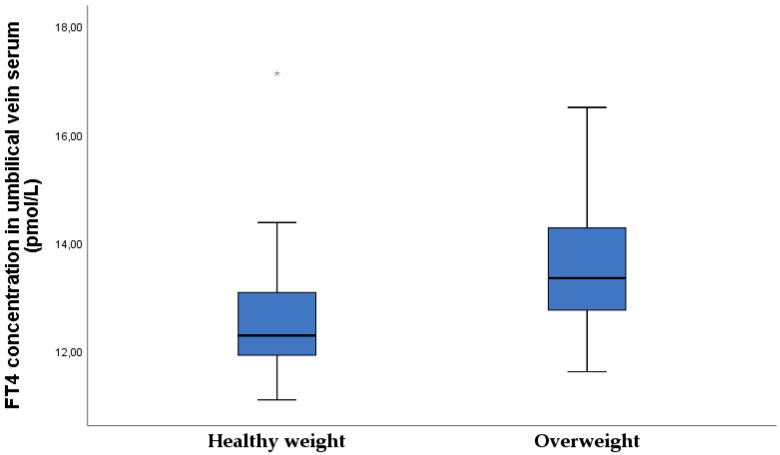
FT4 concentration in umbilical vein serum in healthy-weight and overweight neonates (healthy weight 12.6 ± 1.2: overweight 13.5 ± 1.2, *p* = 0.010).

**Table 1 nutrients-15-00600-t001:** Maternal demographic, obstetric, and laboratory data.

Demographic Data	N	Minimum	Maximum	Mean ± SD, *n* (%) or Median and IQR
Age (years)	66	19	38	30.5 ± 5.3
Duration of T1DM	66	2.0	36	14.3 ± 9.4
Less than 8 years	22	2.0	8.0	4.6 ± 2.1
More than 8 years	44	9.0	36	19.1 ± 6.5
Years of onset of T1DM	66	2.0	36	16.2 ± 9.2
Before 10 years	24	2.0	10.0	7.0 ± 2.7
After 10 years	42	11.0	36.0	21.5 ± 7.6
Height (cm)	66	156	180	166.3 ± 5.3
BMI before pregnancy (kg/m^2^)	66	17.3	39.8	23.9 ± 4.8
BMI < 25 *n* (%)	45 (68.2)	18.7	24.6	21.5 ± 1.7
BMI 25–29.9 *n* (%)	13 (19.7)	25.5	29.4	27.1 ± 3.6
BMI > 30 *n* (%)	8 (12.1)	30.4	39.8	33.2 ± 3.0
Obstetrical data				
Nulliparous *n* (%) Multiparous *n* (%)	38 (57.6) 28 (42.4)			
Gestational age at delivery	66	34	39	38.1 ± 0.9
Gestational weight gain (kg)	66	0	22	13.0 ± 4.6
BMI at delivery (kg/m^2^)	66	21.7	43	28.4 ± 4.7
BMI < 25 *n* (%)	17 (25.8)	21.7	24.9	23.3 ± 1.1
BMI 25–29.9 *n* (%)	28 (42.4)	25.3	29.9	27.4 ± 1.5
BMI > 30 *n* (%)	21 (31.8)	30.0	43.0	33.9 ± 3.5
Maternal laboratory data				
HbA1c (%) in first trimester	66	5.4	8.0	6.8±1.1 ^a^
HbA1c ≤ 6.5 *n* (%) first trimester	32 (48.5)			
HbA1c ≥ 6.6 *n* (%) first trimester	34 (51.5)			
HbA1c (%) in second trimester	65	4.6	8.5	6.3 ± 0.9
HbA1c ≤ 6.5 *n* (%) second trimester	47 (72.3)			
HbA1c ≥ 6.6 *n* (%) second trimester	18 (27.3)			
HbA1c (%) in third trimester	65	4.2	8.6	6.0 ± 0.9 ^a^
HbA1c ≤ 6.5 *n* (%) third trimester	37 (56.9)			
HbA1c ≥ 6.6 *n* (%) third trimester	28 (43.1)			
TSH in first trimester (mIU/L)	63	0.02	5.0	2.5 ± 1.2 2.5 (1.7–3.1)
Subclinical hypothyroidism NO *n* (%) YES *n* (%)	31 (49.2) 32 (50.8)			
FT3 in first trimester (pmol/L)	63	0.86	5.5	3.8 ± 0.6
FT4 in first trimester (pmol/L)	63	8.7	15.2	11.7 ± 1.3
Mean glucose in first in capillary plasma (mmol/L)	64	2.9	10.1	5.8 ± 1.4
Mean glucose in second in capillary plasma (mmol/L)	61	3.3	7.9	5.3 ± 1.2
Mean glucose in third in capillary plasma (mmol/L)	62	3.5	7.5	5.5 ± 1.0
Glucose concentration in maternal vein serum at the time of delivery (mmol/L)	63	2.0	10.2	5.4 ± 1.8 5.2 (4.0–6.2)
CRP (mg/L) in first trimester (mg/L)	54	0.1	19	2.3 (1.1–4.8) ^b^
CRP (mg/L) in second trimester	34	0.2	54	2.9 (1.6–8.3)
CRP (mg/L) in third trimester	35	0.3	34	3.5 (1.7–9.6) ^b^
BDNF in first trimester (ng/L)	61	377.9	2472.0	779.4 (571.9–1055.6) ^c^
BDNF in second trimester (ng/L)	61	213.2	1783.0	632.8 (502.6–803.0)
BDNF in third trimester (ng/L)	54	24.3	1887.0	619.8 (474.4–999.0)
BDNF at the time of delivery (ng/L)	60	19.9	2558.0	592.3 (419.0–908.9) ^c^
Leptin in first trimester (ng/L)	58	3.2	119.7	14.7 (9.6–25.7) ^d^
Leptin in second trimester (ng/L)	62	3.7	150.0	22.6 (13.8–34.6)
Leptin in third trimester (ng/L)	58	2.1	131.5	18.6 (11.6–39.1)
Leptin at the time of delivery (ng/L)	60	1.7	150.0	21.0 (11.8–43.5) ^d^

The number, lowest, highest, and mean values with standard deviation or median and interquartile range (IQR), Wilcoxon sign test ^a,c,d^
*p* < 0.001, statistically significant at Tukey’s corrected α < 0.05; Wilcoxon sign test ^b^
*p* < 0.01, statistically nonsignificant at Tukey’s corrected α < 0.05.

**Table 2 nutrients-15-00600-t002:** Neonatal clinical characteristics and laboratory data.

Neonatal Data	N (%)	Minimum	Maximum	Mean ± SD or Median and IQR
Male *n* (%) Female *n* (%)	29 (43.9) 37 (56.1)			
Term birth *n* (%) Preterm birth *n* (%)	59 (89.4) 7 (10.6)			
Birthweight (g)	66	2500.0	4840	3614.7 ± 584.0
Length (cm)	66	45	54	49.6 ± 2.3
Z-score neonatal weight	66	−1.0	3.2	0.9 (0.4-1.6)
Percentile of neonatal z-score	66	15.2	99.9	76.4 ± 21.0
Healthy-weight neonate *n* (%)	42 (63.6%)	2500	4420	3414.8 ± 550.2
Overweight neonate *n* (%)	24 (36.4%)	3230	4840	3964 ± 472.9
Neonatal macrosomia (≥4000 g) YES *n* (%) NO *n* (%)	17 (25.8) 49 (74.2)			
LGA (≥90 percentile) YES *n* (%) NO (%)	47 (71.2) 19 (28.8)			
Head circumference (cm)	66	31.5	43.5	34.8 ± 1.5
Abdominal circumference (cm)	66	25	38	33.3 ± 2.7
Apgar index at 1 min	66	8	10	9.88 ± 0.37
Apgar index at 5 min	66	9	10	9.95 ± 0.21
Neonatal skinfold thickness on third day after delivery				
Subscapular (cm)	66	3.2	8.6	5.6 ± 1.2
Abdomen (cm)	66	2.4	8.5	4.0 ± 1.0
Biceps (cm)	66	2.5	6.2	4.0 ± 0.8
Triceps (cm)	66	3.1	7.6	5.2 ± 1.0
Femur (cm)	66	3.9	10.4	6.8 ± 1.4
Sum of skinfold thicknesses (cm)	66	16.1	35.9	25.6 ± 3.8
Fat mass percentage (%)	66	8.0	17.9	12.6 ± 1.8
Fat body mass (g)	66	200.8	780.7	459.3 ± 119.8
Lean body mass (g)	66	2299.2	4087.6	3155.4 ± 484.2
Neonatal laboratory data				
pH in umbilical vein blood	38	6.90	7.36	7.21 ± 0.08
TSH in umbilical vein (mIU/L)	59	1.44	14.52	5.6 ± 2.9 4.9 (3.7–6.6)
FT3 in umbilical vein (pmol/L)	52	1.64	4.21	2.6 ± 0.6
FT4 in umbilical vein (pmol/L)	50	11.09	17.13	13.0 ± 1.3
Glucose concentration in umbilical vein serum (mmol/L)	60	1.1	11.6	4.6 ± 2.0 4.5 (3.1–5.3)
The ratio between maternal and umbilical vein serum glucose concentration (mmol/L)	58	0.65	2.41	1.25 ± 0.37 1.2 (1.0–1.5)
C-peptide in umbilical vein serum (pmol/L)	53	37	1015	820 (570–1380)
IR HOMA 2 in umbilical vein serum	53	0.5	7.0	1.7 (1.1–3.2)
BDNF in umbilical vein serum (ng/L)	60	81.8	903.3	304.7 (231.6–458.3)
Leptin in umbilical vein serum (ng/L)	60	0.3	125.5	13.5 (8.9–27.1)

**Table 3 nutrients-15-00600-t003:** Maternal demographics, obstetric, and laboratory data of two study groups.

Variable	*n*	Healthy-Weight Neonates	*n*	Overweight Neonates	*p*
Age (years)	42	31.4 ± 4.8	24	28.8 ± 5.7	0.049
<30 years	14	33.3%	12	50.0%	0.202 ^#^
>30 years	28	66.7%	12	50.0%	
Duration of T1DM	42	13.9 ± 9.8	24	15.0 ± 7.6	0.652
Less than 8 years	16	38.1%	5	21.7%	0.172 ^#^
More than 8 years	24	61.9%	18	78.3%	
Years of onset of T1DM (years)	42	15 (10–26)	24	12 (8.5–17.0)	0.142
Before 10 years *n*%	13	31.0%	11	45.8%	0.290 ^#^
After 10 years *n*%	29	69.0%	13	54.2%	
Height (cm)	42	166.9 ± 5.6	24	165.3 ± 4.7	0.253
BMI before pregnancy (kg/m^2^)	42	23.6 ± 5.0	24	24.4 ± 4.4	0.498
BMI < 25 *n* (%)	31	73.8%	14	58.3%	0.117 ^#^
BMI 25–29.9 *n* (%)	5	11.9%	8	33.3%	
BMI ≥ 30 *n* (%)	6	14.3%	2	8.3%	
Obstetrical data					
Primiparous *n* (%)	26	61.9%	12	50.0%	0.439 ^#^
Multiparous *n* (%)	16	38.1%	12	50.0%	
Gestational age at delivery	42	38.0 ± 1.0	24	38.1 ± 0.6	0.719
Gestational weight gain (kg)	42	12.4 ± 4.7	24	14.1 ± 4.2	0.151
HbA1c first trimester (%)	42	6.6 ± 1.0	24	7.1 ± 1.3	0.073
HbA1c 1. trimester ≤6.5 (%)	25	59.5%	7	29.2%	0.023 ^#^
HbA1c 1. trimester ≥6.6 (%)	17	40.5%	17	70.9%	
Hypothyroidism NO	19	47.5%	12	52.2%	0.797 ^#^
Hypothyroidism YES	21	52.5%	11	47.8%	
TSH in first trimester (mIU/L)	39	2.2 (1.4–3.0)	23	2.7 (1.7–3.9)	0.255
FT3 in first trimester	39	3.8 ± 0.5	23	3.9 ± 0.8	0.536
FT4 in first trimester	39	11.9 ± 1.3	23	11.5 ± 1.2	0.248
BDNF first trimester (ng/L)	40	752.8 (551.1–953.9)	21	850.4 (626.4–1199.0)	0.139
Leptin first trimester (µg/L)	39	15.1 (10.5–29.8)	19	12.5 (8.5–25.7)	0.278

^#^*p* value calculated with χ^2^ test.

**Table 4 nutrients-15-00600-t004:** Neonatal and laboratory data of two study groups.

Variable	*n*	Healthy-Weight Neonates	*n*	Overweight Neonates	*p*
Male *n* (%)	17	40.5%	12	50.0%	0.697 ^#^
Female *n* (%)	25	59.5%	12	50.0%	
Term birth *n* (%)	36	85.7%	23	95.8%	0.408 ^#^
Preterm birth *n* (%)	6	14.3%	1	4.2%	
Birthweight (g)	43	3414.8 ± 550.2	23	3964.6 ± 472.9	<0.001
Length (cm)	43	49.5 ± 2.5	23	49.8 ± 1.8	0.670
Z-score	42	0.6 (0.1–0.8)	24	1.8 (1.6–2.3)	<0.001
Percentile (%)	42	65.2 ± 18.5	24	99.1 ± 1.1	<0.001
Head circumference (cm)	42	34.6 ± 1.8	24	35.2 ± 1.0	0.117
Abdominal circumference (cm)	42	32.8 ± 2.8	24	34.2 ± 2.1	0.038
Apgar index at 1 min	42	9.9 ± 0.3	24	9.9 ± 0.2	0.458
Apgar index at 5 min	42	9.8 ± 0.5	24	9.9 ± 0.3	0.342
pH in umbilical vein blood	25	7.19 ± 0.09	13	7.25 ± 0.04	0.052
Skinfold thickness					
Subscapular (cm)	42	5.3 ± 0.9	24	6.3 ± 1.3	<0.001
Abdomen (cm)	42	3.8 ± 0.8	24	4.5 ± 1.1	0.006
Biceps (cm)	42	3.8 ± 0.8	24	44 ± 0.9	0.008
Triceps (cm)	42	4.9 ± 0.9	24	5.5 ± 0.9	0.011
Femur (cm)	42	6.4 ± 1.2	24	7.5 ± 1.4	0.002
Sum skinfold thickness (cm)	42	24.2 ± 3.4	24	28.2 ± 3.3	<0.001
Fat body mass percentage (%)	42	12.0 ± 1.6	24	13.6 ± 1.7	<0.001
Fat body mass (g)	42	410.7 ± 91.5	24	544.4 ± 117.4	<0.001
Lean body mass (g)	42	3004.1 ± 477.2	24	3420.2 ± 378.2	<0.001
Umbilical vein serum laboratory data					
TSH in the umbilical vein	37	5.1 (3.7–6.2)	22	4.8 (3.9–6.6)	0.913
FT3 in umbilical vein	33	2.4 (2.2–2.7)	19	2.8 (2.1–3.3)	0.093
FT4 in umbilical vein	31	12.3 (11.9–13.2)	19	13.3 (12.7–14.5)	0.002
Glucose concentration in umbilical vein serum (mmol/L)	35	4.6 (3.2–5.9)	24	4.0 (3.5–4.9)	0.254
Maternal–umbilical vein serum glucose ratio concentration (mmol/L)	34	1.0 (0.9–1.4)	24	1.3 (1.1–1.5)	0.013
C-peptide in umbilical vein serum (nmol/L)	35	0.78 (0.6–1.2)	18	0.9 (0.5–1.6)	0.735
The ratio between glucose and C-peptide in umbilical vein serum	31	5.3 (4.2–7.2)	18	3.9 (2.9–5.9)	0.062
IR HOMA 2 in umbilical vein	35	1.7 (1.4–2.6)	18	1.8 (1.0–3.2)	0.985
BDNF in umbilical vein serum (ng/L)	39	303.6 (253.2–388.2)	21	433.0 (230.0–528.0)	0.443
Leptin in umbilical vein serum (ng/L)	38	13.5 (8.5–22.9)	22	13.2 (9.4–29.2)	0.510

^#^*p* value calculated with χ^2^ test.

**Table 5 nutrients-15-00600-t005:** Odds of the overweight neonate for BMI, HbA1c, leptin, and BDNF in the first trimesters of pregnancy.

First Trimester of Pregnancy			
	OR	95% CI	*p* Value
Age of patients (y)	0.875	0.717–1.067	0.186
Duration of type-1 diabetes (y)	0.905	0.765–1.070	0.242
Primiparous/multiparous	0.227	0.031–1674	0.146
HbA1c (%) in first trimester	1.146	0.571–2.300	0.702
BMI before pregnancy	2.171	1.120–4.209	0.022
Leptin in first trimester	0.760	0.603–0.957	0.020
BDNF in first trimester	1.003	1.001–1.006	0.025
TSH in first trimester	2.467	1007–6.004	0.048

## Data Availability

The data presented in this study are available on request from the corresponding author Josip Delmis.

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
