# Peer review of "Relationship of Glucose, C-peptide, Leptin, and BDNF in Maternal and Umbilical Vein Blood in Type-1 Diabetes"

_nutrients, 2023, doi:10.3390/nu15030600_

Round 1
Reviewer 1 Report
The paper “Relationship of Glucose, C-peptide, Leptin, and BDNF in Maternal and Umbilical vein Blood in Type-1 Diabetes” aimed to determine the relationship between glucose, C-peptide, BDNF, and leptin between mother and fetus and neonatal weight, these results are very interesting with sufficient experimental data and in line with readers' interests of nutrients. However, there are still some shortcomings that need to be improved or explained.
Comments:
Section 1: Abstract
Q1. All abbreviations were suggested to supplement full names when they appeared for the first time.
Section 2: Introduction
Q2. Is the umbilical vein Blood mother's blood or the baby's blood?
Q3. “Type 1 diabetes (T1DM) is an autoimmune disease due to selective β-cell destruction and insulin secretion defects”, did these expressions indicate that people suffering T1DM are not prone to obesity? Then why did the authors analyze “Overweight neonates”. Are there any close correlation?
Q4. The last paragraph did also not well present the research significance and innovation, which is suggested to be improved.
Section 3: Results
Q5. Table 1, BMI 25-29.9 n (%), the Minimum value was 21.3? Please check or explain. Similar confusions appear in the next table, Table 2, Overweight neonate min 3230 g, while the Healthy weight neonate max 4420 g.
Q6. Did the authors determined these indicators with significant differences in babies’ bloods? Are any babies suffering T1DM?
These data are sufficient and significative, and I have no other any questions.
Author Response
The paper "Relationship of Glucose, C-peptide, Leptin, and BDNF in Maternal and Umbilical vein Blood in Type-1 Diabetes" aimed to determine the relationship between glucose, C-peptide, BDNF, and leptin between mother and fetus and neonatal weight, these results are very interesting with sufficient experimental data and in line with readers' interests of nutrients. However, some shortcomings need to be improved or explained.
Comments:
Section 1: Abstract
Q1. All abbreviations were suggested to supplement full names when they appeared for the first time.
- It is corrected as suggested.
- BDNF Brain-derived neurotrophic factor
- T1DM Type-1 diabetes mellitus
- CRP C reactive protein
- TSH Thyroid-stimulating hormone FT3-free triiodothyronine
- FT4 free thyroxine
Section 2: Introduction
Q2. Is the umbilical vein Blood mother's blood or the baby's blood?
- Umbilical vein blood is fetal blood. There is a difference between the concentration of AA, FA, Hb, and glucose between the umbilical vein and the umbilical artery. (Holm MB et al. Uptake and release of amino acids in the fetal placental unit in human pregnancies. PLoS ONE. 2017;12:e0185760. Ivanisevic M, et al. Concentrations of fatty acids among macrosomic neonates delivered by healthy women and women with type 1 diabetes mellitus. Int J Gynaecol Obstet. 2020J;149(3):309-317.) we have taken blood from the umbilical vein and not from both the vein and artery (cord blood). The blood in the umbilical vein is well-oxygenated and contains a higher glucose concentration than the umbilical artery.
Q3. "Type 1 diabetes (T1DM) is an autoimmune disease due to selective β-cell destruction and insulin secretion defects" did these expressions indicate that people suffering from T1DM are not prone to obesity? Then why did the authors analyze "Overweight neonates"? Is there any close correlation?
- Patients with T1DM lack endogenous insulin, so they must take exogenous insulin. Patients with T1DM may be obese. A higher intake of nutrients also requires a higher dose of insulin, which can lead to fat tissue accumulation and obesity.
- • The term overweight neonates is defined according to Birth weight z-score specific for sex and gestational age were calculated according to the growth curves published by WHO in 2006 (World Health Organization. WHO multicentre growth reference study: WHO child growth standards: length/height-for-age, weight-for-age, weight-for lengths, weight-for-height and body mass index-for-age: Methods and development. Geneva: WHO 2002. https://www.who.int/publications/i/item/924154693X). The z-score for neonatal weight provides information on underweight (percentile <2), healthy-weight (percentile ≥2 < 98), and overweight (≥ 98) neonates. In type 1 diabetes, there is no correlation between maternal BMI and birth weight.
Q4. The last paragraph also did not present the research significance and innovation, which is suggested to be improved.
- So far, no research has been published on the relationship between glucose, C-peptide, BDNF, and leptin between the mother and the umbilical vein blood in T1DM pregnant women during cesarean section and the association between fetal adipose tissue and leptin and T4. The study aimed to determine the relationship between glucose, C-peptide, BDNF, and leptin between mother and fetus. The secondary endpoint of the research was to analyze the impact of maternal age, duration of type 1 diabetes mellitus, BMI, HbA1c, TSH, leptin, and BDNF on neonatal overweight.
Section 3: Results
Q5. Table 1, BMI 25-29.9 n (%), the Minimum value was 21.3. Please check or explain.
- Thank you, we corrected it. BMI 25-29.9 n (%), the minimum value in our study was 25.5.
Q 5a. Similar confusions appear in the next table, Table 2; overweight neonate min 3230 g, while the Healthy weight neonate max 4420 g.
- It is not a mistake; newborns can have a larger body mass, but the correction of their growth and inclusion in normal newborns then depends on their gender and body length at birth, as well as the gestational week in which they were born. For this purpose, the World Health Organization (WHO) growth curves are used. Birth weight z-scores specific for sex and gestational age was calculated according to the growth curves published by the WHO in 2006 [World Health Organization. WHO Multicentre Growth Reference Study: WHO Child Growth StandardsThe WHO Child Growth Standards: Length/Height-for-Age, Weight-for-Age, Weight-for-Lengths, Weight-for-Height, and Body Mass Index-for-Age: Methods and Development. Geneva: WHO, 2002. https://www.who.int/publications/cra/chapters/volume1/0302-0314.pdf The Z-score for neonatal weight provides information on underweight (percentile <2), healthy-weight (percentile ≥2 and <98), and overweight (percentile ≥98) neonates.
Q6. Did the authors determine these indicators with significant differences in babies' blood?
Are any babies suffering from T1DM?
- We showed differences in the concentration of C-peptide, BDNF, leptin, and T4 between overweight and healthy-weight newborns. It is expected that about 2% of children will develop T1DM during childhood, and if both parents have T1DM, the probability of T1DM is 20%.
These data are sufficient and significant, and I have no other questions.
Reviewer 2 Report
The study aimed to determine the relationship between glucose, C-peptide, BDNF, and 12 leptin between mother and fetus and neonatal weight.
The findings of the study are on line with previous research and not completely new. As pointed out from the authors, the main limit is the lacking control healthy group. Possibly, the reselts could be compared to what is considered "normal" for the tested outcome.
Minor comments:
Line 54: ponderal index (without P)
Line 72: deposit
Line 77: at what time point in life?
Line 404: "The authors speculate that newborns of diabetic mothers have leptin resistance, which leads to the inevitable metabolic repercussions later in life. That is why strict glycemic control during pregnancy 406 is essential"
From which evidence stem such conclusion? Please explain.
Author Response
The study aimed to determine the relationship between glucose, C-peptide, BDNF, and leptin between mother and fetus and neonatal weight.
The findings of the study are in line with previous research and not completely new. As pointed out by the authors, the main limit is the lacking control of a healthy group. Possibly, the results could be compared to what is considered "normal" for the tested outcome.
Minor comments:
Line 54: ponderal index (without P)
- It was corrected.
Line 72. Deposit.
- McLaughlin T, et al. Preferential fat deposition in subcutaneous versus visceral depots is associated with insulin sensitivity. J Clin Endocrinol Metab. 2011 Nov;96(11):E1756-60.) distinguish between the terms depot and deposit. The authors recommend the term depot and subcutaneous deposit for visceral adipose tissue. We have shown the total fetal mass of adipose tissue in the results, so we think the term depot would be more appropriate. Let the Editor of the journal Nutrients decide which term is more appropriate, depot and deposit.
Line 77: at what time point in life?
- Women with type 1 diabetes mellitus can develop subclinical hypothyroidism at any point. During reproductive age, if they develop this type of thyroid dysfunction, it can be reflected in their fertility and cause multiple pregnancy complications, addressed in this study.
Line 404: "The authors speculate that newborns of diabetic mothers have leptin resistance, which leads to the inevitable metabolic repercussions later in life. That is why strict glycemic control during pregnancy is essential."
From which evidence stem such a conclusion? Please explain.
- According to the reference: Higgins MF et al. [34] found significantly higher values of fetal leptin in pregnant women with T1DM and T2DM compared to non-diabetic mothers. The authors [34] speculate that newborns of diabetic mothers have leptin resistance, which leads to the inevitable metabolic repercussions later in life. That is why strict glycemic control during pregnancy is essential. C-peptide and leptin levels were higher in overweight than in eutrophic newborns.